# Decreased SIRT1 Activity Is Involved in the Acute Injury Response of Chondrocytes to Ex Vivo Injurious Mechanical Overload

**DOI:** 10.3390/ijms24076521

**Published:** 2023-03-30

**Authors:** Sonali Karnik, Hessam Noori-Dokht, Taylor Williams, Amin Joukar, Stephen B. Trippel, Uma Sankar, Diane R. Wagner

**Affiliations:** 1Department of Mechanical and Energy Engineering, Indiana University-Purdue University Indianapolis, Indianapolis, IN 46202, USA; 2Department of Orthopaedic Surgery, Indiana University School of Medicine, Indianapolis, IN 46202, USA; 3School of Mechanical Engineering, Purdue University, West Lafayette, IN 47907, USA; 4Department of Biomedical Engineering, Indiana University-Purdue University Indianapolis, Indianapolis, IN 46202, USA; 5Department of Anatomy, Cell Biology and Physiology, Indiana University School of Medicine, Indianapolis, IN 46202, USA

**Keywords:** chondrocyte injury, SIRT1, mechanobiology, impact load, post-traumatic osteoarthritis

## Abstract

A better understanding of molecular events following cartilage injury is required to develop treatments that prevent or delay the onset of trauma-induced osteoarthritis. In this study, alterations to SIRT1 activity in bovine articular cartilage explants were evaluated in the 24 h following a mechanical overload, and the effect of pharmacological SIRT1 activator SRT1720 on acute chondrocyte injury was assessed. SIRT1 enzymatic activity decreased as early as 5 min following the mechanical overload, and remained suppressed for at least 24 h. The chondrocyte injury response, including apoptosis, oxidative stress, secretion of inflammatory mediators, and alterations in cartilage matrix expression, was prevented with pharmacological activation of SIRT1 in a dose-dependent manner. Overall, the results implicate SIRT1 deactivation as a key molecular event in chondrocyte injury following a mechanical impact overload. As decreased SIRT1 signaling is associated with advanced age, these findings suggest that downregulated SIRT1 activity may be common to both age-related and injury-induced osteoarthritis.

## 1. Introduction

Advanced age and previous joint injury are two well-established risk factors for osteoarthritis (OA), a debilitating disease of articular joints characterized by joint pain and cartilage degeneration. Age is by far the most common risk factor for the development of OA, with OA in the knee increasing in prevalence to 50% for those 80+ years of age [1]. Individuals subjected to joint injury are also predisposed to OA, as injury to the knee joint is associated with a 4- to 7-fold increase in the risk of knee OA in young adults [2,3,4,5,6]. Although the progression of post-traumatic OA (PTOA) is complex and is not fully understood, mechanical overload to cartilage produces a cellular response that contributes to disease progression [7,8]. A single traumatic cartilage overload generates a chondrocyte injury response that includes apoptosis, oxidative stress, release of inflammatory mediators, and alterations in gene expression [9,10,11,12,13]. The specific signaling pathways that initiate age-related and post-traumatic OA are not well defined but are generally assumed to be distinct. However, overlap in some cellular behaviors between the two forms of the disease has been noted [14]. Additionally, injury-induced OA has a similar clinical presentation to age-related OA at later stages of the disease, and both lack effective disease-modifying therapies that delay or prevent the progression of joint dysfunction. A better understanding of the similarities and differences in regulation of OA arising from advanced age and joint injury could provide insight into the mechanisms of cartilage health and disease.

Sirtuin1 (SIRT1), a class III protein deacetylase, is a pivotal regulator of aging and longevity. Its overexpression extends the lifespan of several organisms [15,16,17,18,19,20]. SIRT1 activity and expression decrease with advanced age, and decreased SIRT1 activity is considered a hallmark of aging at the cellular level [21,22]. SIRT1 deacetylates various substrates to regulate many aspects of cell behavior, including apoptosis, oxidative stress, inflammation, energy metabolism, and DNA repair [23,24,25]. Low SIRT1 expression and activity in chondrocytes are also associated with OA. SIRT1 knockout in chondrocytes led to accelerated development of OA in surgically-induced joint instability in mice [26,27]. Additionally, genetic or pharmacologic activation of SIRT1 reverses deleterious effects of joint disease. For instance, overexpression of SIRT1 inhibited OA-associated gene expression in human chondrocytes [28]. Similarly, direct and indirect pharmacological stimulation of SIRT1 delayed or prevented OA in animal models [26,29,30,31,32,33].

Collectively, data from the literature suggest that the age-related decline in SIRT1 enzymatic activity in chondrocytes contributes to the increased prevalence of OA with advanced age. Although alterations in SIRT1 activity following a mechanical cartilage overload have not been reported, several aspects of chondrocyte injury following a traumatic overload are consistent with suppressed SIRT1 signaling. These include increased apoptosis and oxidative stress, decreased expression of cartilage matrix components, and the release of inflammatory markers. Therefore, the overall objective of this study was to determine whether regulation of SIRT1 is involved in the acute injury response to a mechanical cartilage overload. To this end, we first developed an injury model to deliver an impact to bovine cartilage explants in order to investigate the acute injury response to mechanical trauma in viable chondrocytes. With this model, alterations to SIRT1 activity were evaluated in the 24 h period following a mechanical overload. The dose-dependent effect of pharmacologic SIRT1 activation on the acute chondrocyte injury response was assessed with SRT1720, a specific and potent activator of SIRT1 that acts via a direct allosteric mechanism [34].

## 2. Results

### 2.1. Sublethal Cartilage Impact Is a Platform to Study Molecular Events in the Chondrocyte Injury Response

A cartilage impact model that was sublethal for at least 24 h was developed so that acute molecular events regulating the injury response to a mechanical overload could be studied without the confounding effects of chondrocyte death. A mechanical impact was delivered to bovine osteochondral explants, with a drop tower that was instrumented with a load cell and accelerometer (Figure 1A). Osteochondral specimens were positioned with the articular surface facing downward on a polished stainless-steel surface, and the bone was impacted from above. The drop tower carriage was released under the force of gravity from a predetermined height. An impact with 12.1 ± 2.0 MPa peak stress, 1.75 ± 0.22 ms duration, and energy density of 1.92 ± 0.65 mJ/mm^3^ was sublethal for 24 h via Live/Dead stain (Figure 1B). An analysis of confocal images indicated that chondrocyte viability at 2 and 24 h post-impact was greater than 98% in all explants, and no difference in chondrocyte viability was detected between impacted and non-impacted groups (Figure 1C). This cartilage impact model was used for all subsequent studies.

### 2.2. SIRT1 Enzyme Activity Decreases with Cartilage Impact and Is Blocked by SIRT1 Activator SRT1720

Decreased SIRT1 activity is observed with advanced age and is associated with exacerbated progression of OA [21,22,26,27]. However, alterations in SIRT1 activity due to cartilage injury had not previously been investigated. SIRT1 enzymatic activity was assessed in chondrocyte lysates in the 24 h period following a mechanical impact overload of cartilage explants. SIRT1 activity decreased more than 30% as early as 5 min post-impact and remained suppressed for 24 h (Figure 2A). *SIRT1* mRNA expression was suppressed 24 h post-impact (Figure 2B), suggesting that decreased SIRT1 activity at later time points may be due, at least in part, to altered gene expression. The ability of pharmacological agent SRT1720 to activate SIRT1 following chondrocyte injury was measured by adding various doses of SRT1720 to explant culture media 30 min before the ex vivo impact. A dose-dependent response to SRT1720 was observed. A concentration of 0.25 µM SRT1720 did not significantly increase SIRT1 enzymatic activity at either 5 min or 24 h post-impact, while concentrations of 1 and 5 µM SRT1720 prevented the decrease in SIRT1 activity at both time points (Figure 2C,D). These SRT1720 concentrations were used to investigate the role of SIRT1 deactivation in the chondrocyte injury response to a mechanical overload.

### 2.3. The Chondrocyte Injury Response 24 h Following Cartilage Impact Is Prevented with SIRT1 Activation

To determine the involvement of SIRT1 in oxidative stress following an impact injury, cartilage explants were fluorescently stained for ROS. Impacted and non-impacted cartilage explants were treated with SRT1720, beginning 30 min before the mechanical overload or exposed to a vehicle control (DMSO). Confocal image analysis of ROS staining indicated a 7-fold increase in untreated explants 24 h post-impact. The treatment of impacted cartilage with 1 and 5 µM SRT1720 decreased ROS staining to levels that were comparable to non-impacted cartilage (Figure 3A). In addition, activated caspase 3/7, an early marker of apoptosis, was visualized in the cartilage explants by fluorescent staining. Caspase 3/7 staining increased substantially 24 h post-impact in untreated explants compared to non-impacted controls. Treatment with 1 and 5 µM SRT1720 prevented activation of the apoptotic pathway and reduced apoptotic signaling to that of non-impacted specimens (Figure 3B).

Whether SIRT1 deactivation modulates the expression of anabolic and catabolic genes in chondrocytes 24 h after a mechanical impact was determined by RT-PCR. The expression of aggrecan and collagen II mRNA in impacted tissue was reduced by approximately 50% compared to non-impacted explants. This decrease in matrix gene expression was prevented with SRT1720 treatment in a dose-dependent manner (Figure 3C). The expressions of matrix metalloproteinases *MMP3* and *MMP13* both increased with the mechanical overload, and this increased expression was at least partially suppressed with SIRT1 activation with SRT1720 (Figure 3C). The expression of aggrecanases *ADAMTS4* and *ADAMTS5* had differential responses to the impact cartilage load. The expression of *ADAMTS4* increased in impacted chondrocytes, and this response was ameliorated with SRT1720 treatment. In contrast, *ADAMTS5* expression decreased with cartilage impact (Figure 3C).

Proinflammatory mediators PGE_2_ and NO are upregulated in chondrocytes exposed to a single impact overload [35]. To investigate whether the modulation of SIRT1 activity affects the release of these inflammatory markers following a mechanical impact, assays were performed on culture medium collected 24 h post-impact. Impact injury increased the PGE_2_ levels released to the medium compared to the non-impacted controls by approximately 2.2-fold (Figure 3D). Both 1 and 5 µm SRT1720 treatment decreased PGE_2_ levels; however, only 5 µM SRT1720 treatment lowered PGE_2_ levels compared to the untreated group. The impact induced 1.7 times higher NO secretion to the medium over 24 h than from non-impacted controls. SRT1720 treatments at 1 and 5 µM significantly decreased NO levels in the medium of impacted explants to levels that were comparable to those of non-impacted groups (Figure 3E).

## 3. Discussion

An injurious impact overload to cartilage is a significant risk factor for the development of OA. Disease initiation can be pinpointed to this event, providing an opportunity to block the progression of PTOA with early treatment. Currently, there are no therapies that prevent or delay disease progression, due in part to insufficient understanding of molecular events following cartilage injury. This study investigated the role of SIRT1 enzymatic activity in the acute chondrocyte injury response to a mechanical overload. The results indicate that SIRT1 activity decreases following an impact overload to cartilage explants, and that downregulated SIRT1 activity is involved in the acute injury response. As decreased SIRT1 activity is associated with aging, these findings reveal a common signaling pathway between advanced age and cartilage injury.

The loading protocol employed in these studies did not produce a significant decrease in chondrocyte viability (Figure 1C) within 24 h. However, an acute chondrocyte injury response was generated 24 h following the impact, including increased oxidative stress and apoptotic signaling (Figure 3A,B), altered mRNA expression (Figure 3C), and elevated secretion of inflammatory markers (Figure 3D,E). This impact model ensured that acute differences in molecular signaling with the mechanical overload were associated with chondrocyte injury and were not due to a loss of viable cells. SIRT1 enzymatic activity decreased as early as 5 min following the mechanical overload and remained suppressed for at least 24 h (Figure 2A), indicating the rapid mechanical regulation of SIRT1 signaling with cartilage injury. These results may be related to those of a previous study, in which SIRT1 downregulation was reported in chondrocytes exposed to a 6 h cyclic stretch [36].

To understand the relationship between SIRT1 activity and chondrocyte injury, SIRT1 deactivation by impact loading was inhibited with the pharmacological agent SRT1720. Nearly all aspects of the chondrocyte injury response that were investigated 24 h post-impact, including indices of apoptosis, oxidative stress, modulation of anabolic and catabolic gene expression, and release of inflammatory mediators were suppressed with 5 µM SRT1720 treatment. One exception was the expression of *MMP3* mRNA, which remained significantly higher in impacted chondrocytes with 5 µM SRT1720 treatment than in non-impacted controls. The lower concentration of 1 µM SRT1720 also maintained SIRT1 activity at levels comparable to the nonimpacted explants, and prevented many but not all indications of cartilage injury. In particular, the altered expressions of anabolic and catabolic genes were not completely prevented when SIRT1 was activated with 1 µM SRT1720 (Figure 3C). These results indicate that the SIRT1 regulatory pathway is involved in the cartilage injury response, but also suggests possible crosstalk with other molecular signaling pathways.

The aggrecanases *ADAMTS4* and *ADAMTS5* were differentially expressed in response to a mechanical overload; *ADAMTS4* expression was upregulated following impact injury while *ADAMTS5* was downregulated. Both contribute to the degradation of human cartilage matrix under normal and inflammatory conditions [37], and both cleave the aggrecan core protein at the same sites [38]. Their differential expression may suggest that overall activity by aggrecanases is relatively unchanged immediately following cartilage injury. The expression pattern of *ADAMTS5* also suggests the possibility that a repair response is activated by a mechanical overload in addition to an injury response. The distinct regulation of *ADAMTS4* and *ADAMTS5* with mechanical impact may indicate specific molecular signaling events in the chondrocyte response.

Collectively, data from the current study implicate SIRT1 deactivation as a key molecular event in chondrocyte injury following a mechanical impact overload. SIRT1 is deactivated within 5 min of a mechanical overload, indicating the very early involvement of this signaling pathway in the injury response. These data raise the possibility that SIRT1 may serve as a molecular target for the development of early interventions that can alter the progression from acute injury to PTOA. Additionally, as decreased SIRT1 signaling is associated with advanced age, these findings suggest that downregulated SIRT1 activity may be common to both age-related and injury-induced OA. SIRT1 regulation in cartilage injury suggests an interrelationship between the chondrocyte response to mechanical overload and aging at the cellular level, and may have implications regarding altered joint susceptibility to injury as individuals age. The current results lay a foundation for further studies defining the cellular response to mechanical overload to cartilage. A more complete understanding of molecular events mediating the chondrocyte injury response may guide efforts to treat or prevent the development of PTOA.

## 4. Materials and Methods

### 4.1. Tissue Harvest and Impact

Osteochondral cores were extracted from fresh bovine metacarpophalangeal joints (Mooresville Butcher Shop, Mooresville, IN, USA) from skeletally mature animals using a sterile 10 mm coring drill bit under a laminar flow hood using aseptic techniques. No more than two specimens were taken per joint. The osteochondral cores were cultured at 37 °C and 5% CO_2_ in defined medium [39] consisting of DMEM (Corning, Corning, NY, USA) supplemented with 1 mM sodium pyruvate, non-essential amino acids, ITS (Gibco, Waltham, MA, USA), and 50 µg/mL L-ascorbic acid (Sigma Aldrich, St. Louis, MO, USA). In some studies, explants were treated with SRT1720 HCl (AdooQ, Irvine, CA, USA) or DMSO (Sigma-Aldrich) vehicle 30 min prior to impact until the time of tissue harvest.

After 24 h in culture, osteochondral cores were impacted using a previously described drop tower [40]. Specimens were placed into a holder with the articular surface facing downward onto a mirror-polished stainless steel plate under aseptic conditions (Figure 1A). A carriage instrumented with a load cell and accelerometer (both manufactured by Kistler, Winterhur, Switzerland) was released from a height of 5 cm under the force of gravity to impact the explant. Data from the load cell and accelerometer were collected at 100 kHz using a laptop connected to a data acquisition module (National Instruments, Austin, TX, USA) and custom Labview code (National Instruments). Data were filtered and analyzed with a custom MATLAB script (MathWorks, Natick, MA, USA) to determine impact characteristics. Non-impacted specimens were placed in the holder, but the drop tower carriage was not released. Cartilage was separated from the bone and cultured for up to 24 h.

### 4.2. Fluorescent Staining and Confocal Imaging

To determine chondrocyte viability, cartilage specimens were cut in half, stained with Live/Dead reagents (Invitrogen, Waltham, MA, USA) and imaged using confocal fluorescent microscopy (Olympus FV-1000 MPE, Tokyo, Japan). Live (green) and dead (red) cells were counted using Fiji software [41]. Live/Dead staining, imaging, and quantification were conducted at least three times to confirm that the impact was sublethal at 24 h. Reactive oxygen species (ROS) and caspase 3/7 signaling were visualized (CellRox and CellEvent stains, respectively, both Invitrogen). Specimen halves were stained, fixed in 4% paraformaldehyde, and were imaged using confocal microscopy. Mean fluorescence intensity was determined (Fiji software, *n* = 3 explants per group).

### 4.3. SIRT1 Enzyme Activity

SIRT1 enzyme activity was assessed using FLUOR DE LYS^®^ SIRT1 fluorometric assay kit (Enzo Life Sciences, Farmingdale, NY, USA) with flash frozen cartilage. Cartilage was homogenized in 2 mM DTT + 1% Triton-X solution with protease inhibitor cocktail (Sigma-Aldrich) in protease- and DNAse-free water. Homogenized samples were centrifuged at 4 °C and supernatant was collected in pre-chilled tubes. The assay buffer was prepared containing the substrate, NAD^+^, and 5 μM Trichostatin A (TSA; Tocris, Bristol, UK) to inhibit the activity of class I and II histone deacetylases (HDACs), which interfere with the specificity of the assay. Sample and assay buffer were added to the wells of a 96-well plate per manufacturer instructions, and the reaction proceeded at 37 °C for 30 min. Fluorescence was read with an excitation of 360 nm and emission at 460 nm. The results were normalized to DNA content by diluting tissue lysates from the SIRT1 assay with Tris EDTA buffer (1:19) and quantifying double stranded DNA (PicoGreen™, Invitrogen).

### 4.4. RNA Extraction and RT-PCR

Flash frozen cartilage tissue was pulverized using a liquid nitrogen cooled mortar and pestle. Trizol (Invitrogen) RNA extraction from cartilage was performed as described by Le Bleu et al. [42] cDNA was constructed from purified RNA using High-Capacity cDNA Reverse Transcription Kit (Applied Biosystem, Waltham, MA, USA). PowerUp™ SYBR™ Green Master Mix (Applied Biosystem) was used as the fluorescent stain for signal detection on the Bio-Rad CFX96™ Real-Time System (Bio-Rad, Hercules, CA, USA). The expression was calculated using the 2^−ΔΔCT^ method relative to *18s* expression (*n* = 3 explants per group). Primer sequences for all bovine genes including *18s* housekeeping gene are provided in Table 1 [43,44,45,46,47].

### 4.5. PGE_2_ ELISA

Prostaglandin E_2_ (PGE_2_) ELISA (R&D Systems, Minneapolis, MN, USA) was performed on medium collected 24 h post-impact. The collected medium was centrifuged at 12,000× *g* at room temperature to separate out any debris prior to performing the assay per the manufacturer’s instructions.

### 4.6. Total NO Assay

Total Nitric Oxide (NO) assay (R&D Systems) utilizing a Greiss reaction was performed on medium collected 24 h post-impact. The collected medium was filtered through 10 K molecular weight cut-off filters (Amicon Ultra-0.5 mL 10 KDa Centrifugal Filter Unit, Millipore-Sigma, Burlington, MA, USA) prior to performing the assay per manufacturer instructions.

### 4.7. Quantification and Statistical Analysis

All data are represented as mean ± standard deviation. The differences in SIRT1 activity with mechanical overload at various times post-impact were determined with two-sided Student’s t-tests at each time point with the Bonferroni correction for multiple comparisons. Other data were analyzed with one- or two-way ANOVA, as appropriate, using Tukey’s or Holm-Sidak’s post hoc analysis (GraphPad Prism, San Diego, CA, USA). Significance was set at *p* < 0.05.

## Figures and Tables

**Figure 1 ijms-24-06521-f001:**
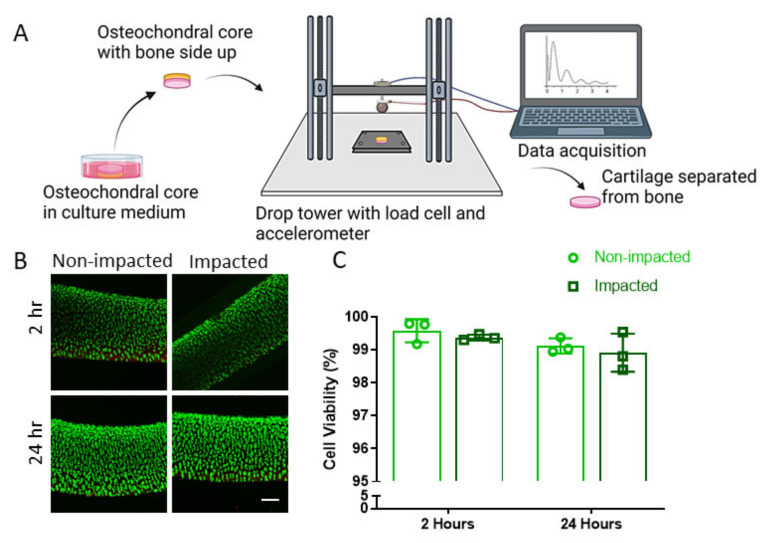
Sublethal cartilage impact is a platform to study acute molecular events in the chondrocyte injury response. (**A**) Schematic of drop tower, impact methods, data acquisition, and post impact tissue processing. (**B**) Live (green) and dead (red) staining of non-impacted and impacted cartilage via confocal microscopy 2 and 24 h post impact. Scale bar = 200 µm. (**C**) Quantification of chondrocyte viability. No significant differences were detected between groups (*n* = 3).

**Figure 2 ijms-24-06521-f002:**
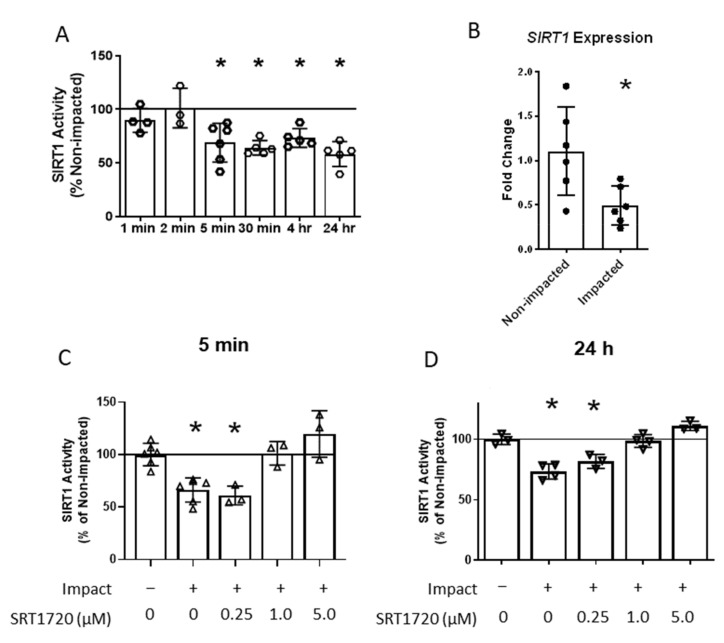
SIRT1 enzyme activity decreases with cartilage impact and is blocked by SIRT1 activator SRT1720. (**A**) SIRT1 enzyme activity at different time points post impact. * indicates statistically significant difference in SIRT1 activity compared to non-impacted cartilage samples at the same time point, uncorrected *p* < 0.05 (*n* = 3 to 5). (**B**) *SIRT1* mRNA expression 24 h post impact relative to *18s* expression. * indicates significant difference from non-impacted group, *p* < 0.05 (*n* = 3). (**C**) SIRT1 enzyme activity 5 min post impact with SRT1720 treatment. * indicates significant difference from non-impacted, untreated controls, *p* < 0.05 (*n* = 5 or 6). (**D**) SIRT1 enzyme activity 24 h post impact with SRT1720 treatment. * indicates significant difference from non-impacted, untreated controls, *p* < 0.05 (*n* = 3 or 4).

**Figure 3 ijms-24-06521-f003:**
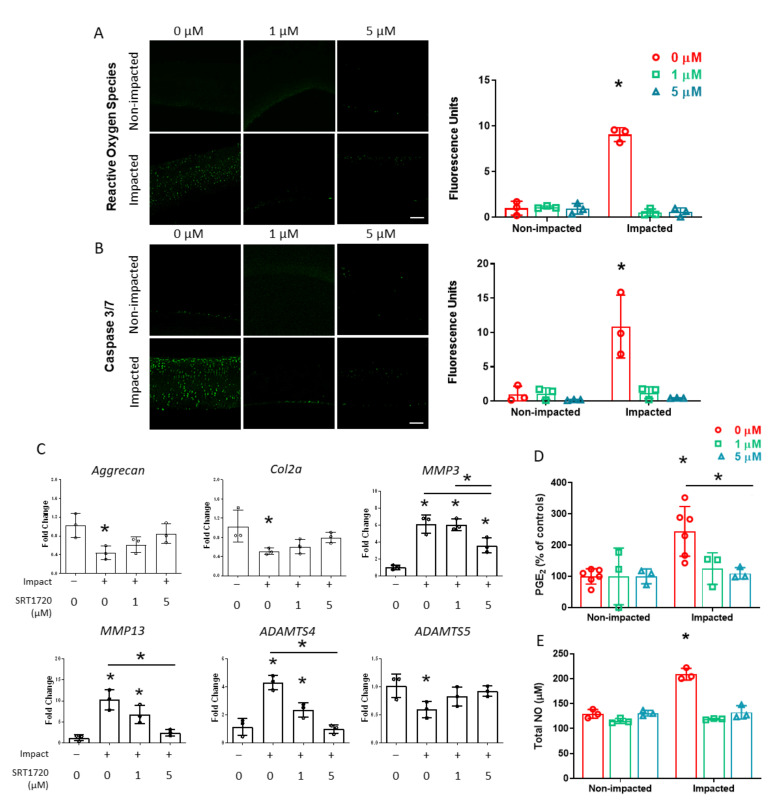
Impact-induced injury response is prevented with pharmacological SIRT1 activation. (**A**) Staining for reactive oxygen species and quantification of mean intensity (*n* = 3). Scale bar = 200 µm. * indicates significant difference from all other groups, *p* < 0.05 (*n* = 3). (**B**) Staining for activated caspase 3/7 and quantification of mean intensity. Scale bar = 200 µm. * indicates significant difference from all other groups, *p* < 0.05 (*n* = 3). (**C**) Anabolic and catabolic mRNA expression 24 h post impact relative to *18s* expression. * over bar indicates significant difference from non-impacted, untreated control group, *p* < 0.05. * over line indicates significant difference between treatment groups in impacted specimens, *p* < 0.05 (*n* = 3). (**D**) PGE_2_ secretion to media 24 h following impact injury. * over bar indicates significant difference from non-impacted, untreated control group, *p* < 0.05. * over line indicates significant difference between treatment groups in impacted specimens, *p* < 0.05 (*n* = 3 or 5). (**E**) NO secretion to media 24 h following impact injury. * indicates statistically significant difference from all other groups, *p* < 0.05 *(n* = 3).

**Table 1 ijms-24-06521-t001:** Primers for PCR experiments.

Gene	Sequence	Reference
*SIRT1*	5′-CAACGGTTTCCATTCGTGTG-3′	[43]
	5′-GTTCGAGGATCTGTGCCAAT-3′	
*Aggrecan*	5′-TGGTGTTTGTGACTCTGAGG-3′	[44]
	5′-GATGAAGTAGCAGGGGATGG-3′	
*Col2a*	5′-AAACCCGAACCCAGAACC-3′	[44]
	5′-AAGTCCGAACTGTGAGAGG-3′	
*MMP3*	5′-TGTGCTCAGCCTATCCACTG-3′	[45]
	5′-AGCTTTCCTGTCACCTCCAA-3′	
*MMP13*	5′-CAATGTTTTTCCTCGAACTCTCAA-3′	[46]
	5′-TTCCACTTCAGAATGAGTCAGATCA-3′	
*ADAMTS4*	5′-GAAGCAATGCACTGGTCTGA-3′	[45]
	5′-CTAGGAGACAGTGCCCGAAG-3′	
*ADAMTS5*	5′-TGCAGATTCTTGCCACAGAC-3′	[45]
	5′-CTTTTGGAGCCGACTTCTTG-3′	
*18s*	5′-GCAATTATTCCCCATGAACG-3′	[47]
	5′-GCCTCACTAAACCATCCAA-3′	

## Data Availability

Not applicable.

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
