# Peer review of "Decreased SIRT1 Activity Is Involved in the Acute Injury Response of Chondrocytes to Ex Vivo Injurious Mechanical Overload"

_ijms, 2023, doi:10.3390/ijms24076521_

Round 1
Reviewer 1 Report
This is a short but interesting work on the role of SIRT 1 in ex vivo mechanical cartilage injury ( bovine cartilage explants).
The work is well conducted. Following a deleteriouis non lethal impact on cartilage explants, the SIRT 1 enzymatic activity decreased and is associated with release of proinflammatory mediators and MMps , and increase in ROS staining as well a decrease in aggrecan and collagen type 2 synthesis. The sublethal impact on cartilage modifications was partially reversed by a pre treatment with SIRT1 720 , an agonist of SIRT 1 . It suggests but not proves a direct relationship between in situ changes in molecules and decrease in sirt1 effect. Several intermediates actors may be involved in a such complex action.
Minor remarks
How the the agonist may penetrate into the normal cartilage and how deep would the SIRT1720 penetrates into the cartilage layers ?
Here , the treatment is purely preventive . What's about a curtaive action of SIRT1 agonist after the deleterious action of this subletahl impact on cartilage?
How authors explain the paradoxical preservation of cell viability. In one hand the cell viabilty was not affected while SIRT 1 activity decrease and caspase and ROS staining increase which all contribute to increase the death of chondrocytes ?
It is not made clear how many times the experiments have been repeated?
Reviewer 2 Report
This succinct work clarified the involvement of impaired SIRT activity in the response of articular chondrocytes after acute injury. Namely, sublethal mechanical impact on chondrocytes in bovine articular cartilage induced ROS production, apoptotic onset, attenuation of anabolic gene expression, induction of catabolic genes and production of inflammatory mediators, which were accompanied by decreased SIRT activity and were ameliorated by a SIRT activator.
Overall, data presented are clear and interesting. Several points need to be addressed, and a few additional experiments would further increase the impact of the present work.
Major points
1) SIRT1 is known to activate a transcription factor, FOXO1, by its enzymatic activity. It is recommended to clarify if FOXO1 is used as a downstream mediator of SIRT1 in the observed response to the mechanical stress.
2) The data in Figure 2B failed to find statistically significant effect of the mechanical impact on SIRT1 mRNA level. However, this may be because of the least sample size (n = 3). If this experiment is repeated, the author may find significant repression of SIRT1 mRNA expression by the mechanical impact.
3) Rather than additional molecular signaling events involved in the unexpected response of ADAMTS5 to mechanical impact, biological significance of this response should be discussed in Discussion section. Since repression of ADAMTS5 was also nullified by the inhibitor, not an additional, but a related molecular signaling event is suspected therein.
Minor points
1) Please describe how live and dead cells are observed in Figure 1B in the legend as well as in Materials and Methods. The sample size (n = 3) should be also given in each legend for the data in each figure.
2) The present indication of statistical significance with "a", "b", "c" and corresponding sentence in the legend "Different letters indicate statistical significance (between the groups)" is not understandable. Please consider another indication.
3) The first letter of Abstract "A" should not be shown in boldface.
4) The first sentence in Discussion may be rephrased. Do the authors mean "an opportunity to block the progression of PTOA"?
5) In Table 1, 5'- and 3'-ends of primer nucleotide sequences should be indicated.
Round 2
Reviewer 2 Report
Most of the issues pointed out have been duly addressed in the revised version.